# Gestational age, birth weight, and perinatal complications in mothers with diabetes and impaired glucose tolerance: Japan Environment and Children's Study cohort

**Hiroshi Yokomichi**[1]*, **Mie Mochizuki**[2], **Ryoji Shinohara**[3], **Megumi Kushima**[3], **Sayaka Horiuchi**[3], **Reiji Kojima**[1], **Tadao Ooka**[1], **Yuka Akiyama**[1], **Kunio Miyake**[1], **Sanae Otawa**[3], **Zentaro Yamagata**[1,3], on behalf of the Japan Environment and Children's Study Group[¶]

**1** Department of Health Sciences, University of Yamanashi, Chuo City, Yamanashi, Japan, **2** Department of Paediatrics, University of Yamanashi, Chuo City, Yamanashi, Japan, **3** Centre for Birth Cohort Studies, University of Yamanashi, Chuo City, Yamanashi, Japan

¶ The members of the Japan Environment and Children's Study Group are listed in the Acknowledgments.
* hyokomichi@yamanashi.ac.jp

**Data Availability Statement:** Data are available on reasonable request. Data are unsuitable for public

## Abstract

We aimed to determine the risk of perinatal complications during delivery in mothers with non-normal glucose tolerance in a large Japanese birth cohort. We analysed data of 24,295 neonate–mother pairs in the Japan Environment and Children's Study cohort between 2011 and 2014. We included 67 mothers with type 1 diabetes, 102 with type 2 diabetes (determined by questionnaire), 2,045 with gestational diabetes (determined by diagnosis), and 2,949 with plasma glucose levels ≥140 mg/dL (shown by a screening test for gestational diabetes). Gestational age, birth weight, placental weight, and proportions of preterm birth, and labour and neonatal complications at delivery in mothers with diabetes were compared with those in mothers with normal glucose tolerance. Mean gestational age was shorter in mothers with any type of diabetes than in mothers without diabetes. Birth weight tended to be heavier in mothers with type 1 diabetes, and placental weight was significantly heavier in mothers with type 1 and gestational diabetes and elevated plasma glucose levels (all p<0.05). The relative risks of any labour complication and any neonatal complication were 1.49 and 2.28 in type 2 diabetes, 1.59 and 1.95 in gestational diabetes, and 1.22 and 1.30 in a positive screening test result (all p<0.05). The relative risks of preterm birth, gestational hypertension, and neonatal jaundice were significantly higher in mothers with types 1 (2.77; 4.07; 2.04) and 2 diabetes (2.65; 5.84; 1.99) and a positive screening test result (1.29; 1.63; 1.12) than in those without diabetes (all p<0.05). In conclusion, placental weight is heavier in mothers with non-normal glucose tolerance. Preterm birth, gestational hypertension, and jaundice are more frequent in mothers with types 1 and 2 diabetes. A positive result in a screening test for gestational diabetes suggests not only a non-normal glucose tolerance, but also a medium (middle-level) risk of perinatal complications.

deposition because of ethical restrictions and the legal framework of Japan. The Act on the Protection of Personal Information (Act No. 57 of 30 May 2003, amendment on 9 September 2015) prohibits public deposition of the data containing personal information. Ethical Guidelines for Medical and Health Research Involving Human Subjects enforced by the Japan Ministry of Education, Culture, Sports, Science and Technology and the Ministry of Health, Labour and Welfare also restrict the open sharing of epidemiological data. All inquiries about access to data should be sent to: jecsen@nies.go.jp. The person responsible for handling enquiries sent to this email address is Shoji F. Nakayama, JECS Program Office, National Institute for Environmental Studies.

**Funding:** This study was funded by the Ministry of the Environment, Japan (to ZY, grant no: none. https://www.env.go.jp/chemi/ceh/). The funder had no role in study design, data collection and analysis, decision to publish, or preparation of the manuscript. The findings and conclusions of this article are solely the responsibility of the authors and do not represent the official views of the above government.

**Competing interests:** The authors have declared that no competing interests exist.

# Introduction

Non-normal glucose tolerance of pregnant women increases the probability of miscarriage, stillbirth, neonatal congenital abnormalities, and complications of labour [1–4]. Congenital abnormalities include those of the heart and cleft, umbilical cord hernia, hypospadias, and chromosomes. Complications of labour include threatened premature delivery, premature rupture of the membranes, placental abruption, gestational hypertension, macrosomia, and intrauterine infection. To prevent these complications, clinicians attempt to control plasma glucose levels of women who expect children.

In Japan, a 50-g oral glucose challenge test (GCT) in the second trimester of pregnancy is recommended for pregnant women without diabetes as screening [5]. If mothers have a positive result in the GCT, they should undergo detailed examinations [6]. In the HAPO study in which a 75-g oral glucose tolerance test was conducted between 24 and 32 weeks' gestation, the extent of hyperglycaemia was linearly associated with the incidence of caesarean section, large-for-gestational-age, and hyperbilirubinaemia [7]. The Japanese 50-g GCT may also predict these complications at delivery.

Recently, control of plasma glucose levels in mothers with diabetes has become stringent [8]. This improved control of diabetes should have reduced the rate of complications in delivery. However, a recent increase in deliveries of mothers with obesity [9], advanced age [10], and elevated plasma glucose levels might have increased the rate of complications. Therefore, this rate needs to be revised for use in clinical practice. Evidence is also scarce on birth weight and complications in neonates of mothers with mild glucose tolerance. We aimed to investigate birth weight, placental weight, and the incidence of complications during delivery. We compared neonates of mothers with diabetes, mothers with moderately elevated plasma glucose levels, and mothers with normal glucose tolerance using data from a large birth cohort in Japan.

# Materials and methods

## Ethics statement

The Japan Environment and Children's Study (JECS) protocol was reviewed and approved by the Ministry of the Environment's Institutional Review Board on Epidemiological Studies and by the Ethics Committees of all participating institutions. The study was performed in accordance with the ethical guidelines and regulations of the Declaration of Helsinki. All participants and parents or guardians of the children provided written informed consent before participating in the study.

## Measures

Details on the JECS project are published elsewhere [11]. Approximately 100,000 expecting mothers who lived in the designated study areas were recruited over 3 years. We included 15 Regional Centres covering 19 prefectures across Japan [12]. We used data of 24,295 neonate–mother pairs where we could identify the history of diabetes and a glucose tolerance test result of mothers who delivered neonates between March 2011 and November 2014. We used the JECS "jecs-ta-20190930-qsn" dataset, which was released in October 2019. This dataset comprises information on demographic factors, lifestyle, socioeconomic status, environmental exposure, medical history, and delivery information obtained from self-administered questionnaires and medical records transcribed by physicians, midwives/nurses, or trained research co-ordinators.

Mothers answered about the experience of diagnosed type 1 diabetes, type 2 diabetes, and other endocrinological disease in the first trimester in the questionnaire. Data of mothers with

endocrinological disease other than diabetes (e.g., Graves' disease or Hashimoto disease) were excluded from the analysis. Obstetricians conducted a 50-g oral GCT for the pregnant participants at the second trimester. The part of the questionnaire regarding the diagnosis of gestational diabetes mellitus (GDM) during pregnancy, gestational age, and perinatal complications was filled in by obstetricians at delivery and by paediatricians in the neonatal unit.

In Japan, GDM or overt diabetes in pregnancy is screened twice in all pregnant women. In the first trimester of pregnancy, random plasma glucose levels are evaluated with a cut-off value of 100 mg/dL or 95 mg/dL (the cut-off value depends on the institution). In the second trimester, the 50-g oral GCT with a cut-off value of 140 mg/dL at 1 hour or random plasma glucose levels with a cut-off value of 100 mg/dL is used for screening [5]. The GCT is usually conducted regardless of the mealtime. These screening tests are used to evaluate the glucose tolerance in pregnant women. If mothers have a positive result in the screening, they should undergo detailed examinations for determining the presence of GDM or overt diabetes in pregnancy [6].

We categorised mothers in the groups of type 1 diabetes, type 2 diabetes, GDM, screened gestational diabetes mellitus (screened GDM), and non-diabetes as follows. First, we identified mothers with type 1 diabetes, type 2 diabetes, or GDM from the questionnaire. Overlapping of the types of diabetes was not permitted. Next, mothers with plasma glucose levels ≥140 mg/dL [6] at 1 hour in the GCT in the second trimester were categorised into the screened GDM group in this study. Although there were also data of GCT results in the third trimester, we did not use these data because the GCT is recommended to be conducted in the second trimester. Mothers without screened GDM, any type of diabetes, endocrinological disease, or plasma glucose levels ≥140 mg/dL at 1 hour in the GCT in the third trimester were categorised into the non-diabetes group.

## Statistical analyses

We showed data as the mean and standard deviation (SD) for gestational age, birth weight, birth height, head circumference, chest circumference, and placental weight by the type of diabetes. We calculated the proportions of preterm birth, post-term birth, caesarean section, foetal position, neonatal and labour complications, and congenital abnormalities. Poisson regression was used to calculate the relative risks of perinatal complications for type 1 diabetes, type 2 diabetes, GDM, and screened GDM compared with those for the non-diabetes group, with adjustment for maternal age and maternal smoking status in the first trimester in pregnancy. We also calculated the means and SDs of subgroups of the mother's age, and term and single births. Additionally, we calculated the proportions of subgroups of the mother's body mass index (BMI). Missing data were not included in the analysis. We conducted Welch's t test or Fisher's exact test for the comparison of variables between mothers with diabetes and those without diabetes. We conducted all statistical analyses using SAS statistical software (version 9.4; SAS Institute, NC, USA). Two-sided p values of <0.05 were considered to indicate a significant difference.

## Results

Table 1 shows data on gestational age and anthropometric measures of neonates by the type of the mothers' diabetes. The groups of type 1 diabetes, type 2 diabetes, GDM, screened GDM, and non-diabetes comprised 67, 102, 2,045, 2,949, and 19,123 neonates, respectively. The mean birth weight in the type 1 diabetes group (3,043 g) was the heaviest among the groups. The mean placental weight in the type 1 diabetes (p = 0.023), GDM (p<0.0001), and screened GDM groups (p = 0.0002) was significantly heavier than that in the non-diabetes group.

**Table 1. Anthropometrics of neonates of mothers with various types of diabetes and those without diabetes.**

| Mean (standard deviation) | Type 1 | Type 2 | Group GDM | Screened GDM | Non-diabetes |
|---|---|---|---|---|---|
| Number | 67 | 102 | 2,045 | 2,949 | 19,132 |
| Mother's age, years | 30.9 (5.6) | 32.6* (4.4) | 33.4* (5.0) | 32.5* (5.0) | 31.0 (5.1) |
| Gestational age, weeks | 36.5* (5.3) | 37.4* (3.6) | 38.3* (2.1) | 38.7* (1.8) | 38.8 (1.7) |
| Birth weight, g | 3,043 (779) | 2,955 (837) | 2,990 (497) | 3,001 (454) | 2,997 (421) |
| Birth height, cm | 47.9 (6.8) | 48.0 (4.8) | 48.7 (2.7) | 48.7 (2.4) | 48.7 (2.4) |
| Head circumference, cm | 33.2 (1.7) | 33.1 (2.6) | 33.3* (1.9) | 33.2 (1.5) | 33.2 (1.6) |
| Chest circumference, cm | 32.1 (2.4) | 31.6 (3.0) | 31.7 (2.0) | 31.7 (1.9) | 31.7 (1.8) |
| Placental weight, g | 604* (145) | 590 (157) | 580* (147) | 572* (133) | 562 (134) |
| Placenta to birth weight ratio | 0.21 (0.13) | 0.21* (0.06) | 0.20* (0.09) | 0.19 (0.05) | 0.19 (0.15) |

Data are mean (standard deviation) or number.

*p<0.05 compared with the non-diabetes group (by t test).

GDM, gestational diabetes.

Table 2 shows the proportions of preterm birth, complications of labour, neonatal complications, and congenital abnormalities. The proportion of preterm birth was significantly higher in the type 1 diabetes (p = 0.0065), type 2 diabetes (p<0.0001), GDM (p<0.0001), and

**Table 2. Perinatal data of neonates of mothers with various types of diabetes and those without diabetes.**

| Number (%) or mean (standard deviation) | Type 1 | Type 2 | Group GDM | Screened GDM | Non-diabetes |
|---|---|---|---|---|---|
| Gestational age at 22–30 weeks | 1/64 (1.5) | 3/100 (3.0)* | 16/2,041 (0.8)* | 15/2,948 (0.5) | 66/19,120 (0.4) |
| Gestational age at 31–36 weeks | 8/64 (12.5)* | 13/100 (13.0)* | 174/2,041 (8.5)* | 168/2,948 (5.7)* | 887/19,120 (4.6) |
| Gestational age at ≥42 weeks | 0/64 (0) | 0/100 (0) | 5/2,041 (0.2) | 5/2,948 (0.2) | 31/19,120 (0.2) |
| Caesarean section | 25/66 (37.9)* | 47/101 (46.5)* | 626/2,039 (30.7)* | 770/2,945 (26.2)* | 4,015/19,066 (21.1) |
| Multiple births | 0/68 (0) | 2/104 (1.9) | 70/2,046 (3.4)* | 83/2,970 (2.8)* | 418/19,196 (2.2) |
| Miscarriage | 2/67 (3.0)* | 2/102 (2.0)* | 3/2,046 (0.2) | 1/2,958 (0.03) | 11/19,160 (0.1) |
| Stillbirth | 1/67 (1.5) | 3/102 (2.9)* | 4/2,046 (0.2) | 5/2,958 (0.2) | 23/19,160 (0.1) |
| Any labour complication | 31/65 (47.7) | 67/102 (65.7)* | 2,028/2,042 (99.3)* | 1,371/2,911 (47.1) | 8,709/18,897 (46.1) |
| Transverse presentation | 0/64 (0) | 1/101 (1.0) | 11/2,002 (0.6)* | 12/2,899 (0.4)* | 37/18,767 (0.2) |
| Breech presentation | 3/64 (4.7) | 6/101 (5.9) | 76/2,002 (3.8) | 106/2,899 (3.7) | 600/18,767 (3.2) |
| Preterm labour | 8/68 (11.8) | 26/104 (25.0) | 454/2,046 (22.2)* | 508/2,970 (17.1) | 3,452/19,196 (18.0) |
| Early rupture of membranes | 10/68 (14.7) | 15/104 (14.4)* | 245/2,046 (12.0)* | 284/2,970 (9.6) | 1,822/19,196 (9.5) |
| Placental abruption | 1/68 (1.5) | 0/104 (0) | 11/2,046 (0.5) | 17/2,970 (0.6) | 102/19,196 (0.5) |
| Gestational hypertension | 6/68 (8.8)* | 13/104 (12.5)* | 114/2,046 (5.6)* | 104/2,970 (3.5)* | 410/19,196 (2.2) |
| Intrauterine infection | 1/68 (1.5) | 3/104 (2.9) | 17/2,046 (0.8) | 23/2,970 (0.8) | 179/19,196 (0.9) |
| Any neonatal complication | 27/68 (39.7)* | 31/104 (29.8)* | 419/2,046 (20.5)* | 448/2,970 (15.1)* | 2,363/19,196 (12.3) |
| Macrosomia | 2/66 (3.0) | 6/102 (5.9)* | 31/2,042 (1.5)* | 37/2,948 (1.3)* | 120/19,117 (0.6) |
| Jaundice with treatment | 14/68 (20.6)* | 21/104 (20.2)* | 249/2,046 (12.2) | 369/2,970 (12.4)* | 2,083/19,196 (10.9) |
| Abnormalities | 5/68 (7.4)* | 2/104 (1.9) | 77/2,046 (3.8)* | 75/2,970 (2.5) | 477/19,196 (2.5) |
| Heart | 1/68 (1.5) | 0/104 (0) | 32/2,046 (1.6)* | 30/2,970 (1.0) | 140/19,196 (0.7) |
| Cleft | 0/68 (0) | 0/104 (0) | 1/2,046 (0.05) | 2/2,970 (0.07) | 17/19,196 (0.09) |
| Umbilical cord hernia | 0/68 (0) | 0/104 (0) | 1/2,046 (0.05) | 0/2,970 (0) | 4/19,196 (0.02) |
| Hypospadias | 0/68 (0) | 0/104 (0) | 1/2,046 (0.05) | 2/2,970 (0.07) | 10/19,196 (0.05) |
| Chromosomal abnormalities | 0/68 (0) | 0/104 (0) | 6/2,046 (0.3) | 10/2,970 (0.3)* | 27/19.196 (0.1) |

Data are mean (standard deviation) or n (%).

*p<0.05 compared with the non-diabetes group (by t test or Fisher's exact test).

GDM, gestational diabetes.

**Table 3. Relative risks of perinatal outcomes of mothers with type 1 diabetes, type 2 diabetes, and GDM.**

| Relative risk | Type 1 | Type 2 | Group GDM | Screened GDM | Non-diabetes |
|---|---|---|---|---|---|
| Preterm birth | 2.77* | 2.65* | 1.57* | 1.29* | Reference |
| Caesarean section | 1.79* | 1.98* | 1.40* | 1.19* | Reference |
| Multiple births | − | 0.80 | 2.12* | 1.35* | Reference |
| Miscarriage | − | − | − | − | Reference |
| Stillbirth | − | − | − | − | Reference |
| Any labour complication | 1.04 | 1.49* | 1.59* | 1.22* | Reference |
| Transverse presentation | − | − | − | − | Reference |
| Breech presentation | 1.45 | 1.59 | 1.59* | 1.08 | Reference |
| Preterm labour | 0.65 | 1.29 | 1.10 | 0.96 | Reference |
| Early rupture of the membranes | 1.71 | 1.37 | 0.69* | 1.07 | Reference |
| Placental abruption | − | − | − | − | Reference |
| Gestational hypertension | 4.07* | 5.84* | 2.00* | 1.63* | Reference |
| Intrauterine infection | − | − | − | − | Reference |
| Any neonatal complication | 3.18* | 2.28* | 1.95* | 1.30* | Reference |
| Macrosomia | − | − | − | − | Reference |
| Jaundice with treatment | 2.04* | 1.99* | 1.02 | 1.12* | Reference |
| Abnormalities | 3.55* | 1.03 | 1.36 | 1.13 | Reference |
| Heart | 2 | 1.19 | 2.44* | 1.54* | Reference |
| Cleft | − | − | − | − | Reference |
| Umbilical cord hernia | − | − | − | − | Reference |
| Hypospadias | − | − | − | − | Reference |
| Chromosomal abnormalities | − | − | − | − | Reference |

The relative risk was calculated by Poisson regression with adjustment for maternal age and maternal smoking status in the first trimester in pregnancy. The "−" symbol indicates that the analysis was not possible.

*$p < 0.05$ compared with the non-diabetes group.

GDM, gestational diabetes.

screened GDM groups (p = 0.0072) than in the non-diabetes group. The proportion of macrosomia tended to be higher in the type 1 diabetes group (p = 0.07), and was significantly higher in the type 2 diabetes (p<0.0001), GDM (p<0.0001), and screened GDM groups (p = 0.0005) than in the non-diabetes group. The proportions of gestational hypertension and jaundice with treatment were also significantly higher in the type 1 diabetes (p = 0.0035; p = 0.017), type 2 diabetes (p<0.0001; p = 0.0064), and screened GDM groups (p<0.0001; p = 0.012) than in the non-diabetes group.

Table 3 shows the relative risks of perinatal outcomes of mothers with any type of diabetes or high plasma glucose levels. The relative risks of preterm birth for type 1 diabetes, type 2 diabetes, GDM, and screened GDM were 2.77, 2.65, 1.57, and 1.29, respectively (all p<0.05). The relative risks of any labour complication and any neonatal complication for type 1 diabetes, type 2 diabetes, GDM, and screened GDM were 1.04 and 3.18, 1.49 and 2.28, 1.59 and 1.95, and 1.22 and 1.30, respectively. The relative risks of gestational hypertension and jaundice with treatment for type 1 diabetes, type 2 diabetes, GDM, and screened GDM were 4.07 and 2.04, 5.84 and 1.99, 2.00 and 1.02, and 1.63 and 1.12, respectively.

Table 4 shows gestational age and anthropometric measures in relation to the maternal age. In mothers aged 20–49 years, birth weight tended to be higher in the type 1 diabetes group than in the non-diabetes group. In mothers aged 20–49 years, placental weight tended to be heavier in any of diabetes and screened GDM groups than in the non-diabetes group.

**Table 4. Perinatal data of neonates of mothers with various types of diabetes in relation to maternal age.**

| Mean (standard deviation) | Type 1 | Type 2 | Group GDM | Screened GDM | Non-diabetes |
|---|---|---|---|---|---|
| **Mothers' age: 14–19 years** | | | | | |
| Number | – | – | 9 | 10 | 188 |
| Gestational age, weeks | – | – | 39.2 (1.4) | 39.2* (1.0) | 39.1 (1.3) |
| Birth weight, g | – | – | 2961 (434) | 3,059 (385) | 2,974 (360) |
| Placental weight, g | – | – | 583 (84) | 527 (111) | 554 (111) |
| Placenta to birth weight ratio | – | – | 0.20 (0.04) | 0.17* (0.02) | 0.19 (0.04) |
| **Mothers' age: 20–29 years** | | | | | |
| Number | 31 | 25 | 462 | 813 | 7,090 |
| Gestational age, weeks | 37.1* (4.7) | 38.2 (2.4) | 38.7* (1.7) | 38.8* (1.6) | 38.9 (1.7) |
| Birth weight, g | 3,065 (754) | 3,041 (756) | 3,048* (437) | 2,989 (429) | 3,000 (407) |
| Placental weight, g | 607 (137) | 603 (144) | 593* (136) | 576* (155) | 563 (122) |
| Placenta to birth weight ratio | 0.20 (0.05) | 0.20 (0.05) | 0.20* (0.06) | 0.20* (0.06) | 0.19 (0.05) |
| **Mothers' age: 30–39 years** | | | | | |
| Number | 30 | 73 | 1,344 | 1,908 | 11,001 |
| Gestational age, weeks | 36.0* (6.2) | 37.1* (4.0) | 38.3* (2.1) | 38.7 (1.8) | 38.7 (1.7) |
| Birth weight, g | 3,029 (845) | 2,923 (880) | 2,984 (502) | 3,012 (451) | 2,998 (427) |
| Placental weight, g | 599 (163) | 583 (165) | 577* (152) | 571* (122) | 561 (129) |
| Placenta to birth weight ratio | 0.23 (0.18) | 0.21 (0.07) | 0.20* (0.10) | 0.19 (0.05) | 0.19 (0.19) |
| **Mothers' age: 40–49 years** | | | | | |
| Number | 5 | 4 | 228 | 218 | 852 |
| Gestational age, weeks | 37.6 (1.1) | 38.3 (2.1) | 38.0* (2.4) | 38.1* (2.5) | 38.6 (1.8) |
| Birth weight, g | 3,181 (584) | 3,000 (589) | 2,918 (558) | 2,940 (555) | 2,962 (456) |
| Placental weight, g | 595 (109) | 626 (91) | 566 (139) | 568 (141) | 565 (251) |
| Placenta to birth weight ratio | 0.19 (0.03) | 0.21 (0.02) | 0.20 (0.07) | 0.20 (0.06) | 0.19 (0.08) |

Data are mean (standard deviation) or number.

*p<0.05 compared with the non-diabetes group (by t test).

GDM, gestational diabetes.

Table 5 shows gestational age and anthropometric measures in term and single births. The mean gestational age was significantly shorter and the mean birth weight was significantly higher in any of the diabetes groups than in the non-diabetes group (all p<0.05). The mean

**Table 5. Perinatal data of neonates of mothers with various types of diabetes in relation to term and single births.**

| Mean (standard deviation) | Type 1 | Type 2 | Group GDM | Screened GDM | Non-diabetes |
|---|---|---|---|---|---|
| Number | 55 | 82 | 1,817 | 2,713 | 17,912 |
| Gestational age, weeks | 38.3* (1.0) | 38.6* (1.2) | 38.8* (1.2) | 39.0* (1.2) | 39.1 (1.1) |
| Birth weight, g | 3,281* (410) | 3,208* (568) | 3,078* (401) | 3,064 (381) | 3,045 (361) |
| Birth height, cm | 49.8* (2.1) | 49.4 (2.2) | 49.1* (2.0) | 49.0 (2.0) | 49.0 (1.9) |
| Head circumference, cm | 33.5 (1.3) | 33.7* (1.4) | 33.4* (1.4) | 33.4 (1.4) | 33.3 (1.4) |
| Chest circumference, cm | 32.7* (1.7) | 32.3 (2.2) | 32.0* (1.6) | 31.9 (1.6) | 31.9 (1.6) |
| Placental weight, g | 629* (123) | 615* (150) | 573* (128) | 567* (118) | 557 (118) |
| Placenta to birth weight ratio | 0.19 (0.04) | 0.19* (0.03) | 0.19* (0.03) | 0.19* (0.03) | 0.18 (0.03) |

Data are mean (standard deviation) or number.

*p<0.05 compared with the non-diabetes group (by t test).

GDM, gestational diabetes.

**Table 6. Perinatal data of neonates of mothers with various types of diabetes in relation to BMI.**

| Number (%) | Type 1 | Type 2 | Group GDM | Screened GDM | Non-diabetes |
|---|---|---|---|---|---|
| **BMI of ≤18.4 kg/m²** | | | | | |
| Gestational age at 22–30 weeks | 1/7 (14.3)* | 0/5 (0) | 3/220 (1.4)* | 2/434 (0.5) | 9/3,395 (0.3) |
| Gestational age at 31–36 weeks | 2/7 (28.6)* | 1/5 (20.0) | 8/220 (3.6) | 22/434 (5.1) | 179/3,395 (5.3) |
| Caesarean section | 3/7 (42.9) | 3/6 (50.0) | 35/221 (15.8) | 86/435 (19.8) | 555/3,411 (16.3) |
| Any labour complication | 5/7 (71.4) | 5/6 (83.3) | 219/221 (99.1)* | 201/435 (46.2) | 1,634/3,411 (47.9) |
| Any neonatal complication | 2/7 (28.6) | 0/6 (0) | 45/221 (20.4)* | 76/435 (17.5)* | 472/3,411 (13.8) |
| Any abnormalities | 2/7 (28.6)* | 0/6 (0) | 6/221 (2.7) | 11/435 (2.5) | 90/3,411 (2.6) |
| **BMI of 18.5–22.9 kg/m²** | | | | | |
| Gestational age at 22–30 weeks | 0/35 (0) | 0/26 (0) | 4/958 (0.4) | 9/1,688 (0.5) | 33/12,065 (0.3) |
| Gestational age at 31–36 weeks | 2/35 (5.7) | 1/26 (3.9) | 77/958 (8.0)* | 90/1,688 (5.3) | 521/12,065 (4.3) |
| Caesarean section | 14/38 (36.8)* | 9/26 (34.6) | 233/960 (24.3)* | 408/1,703 (24.0)* | 2,463/12,107 (20.3) |
| Any labour complication | 15/38 (39.5) | 16/26 (61.5) | 951/960 (99.1)* | 771/1,703 (45.3) | 5,391/12,107 (44.5) |
| Any neonatal complication | 16/38 (42.1)* | 5/26 (19.2) | 172/960 (17.9)* | 234/1,703 (13.7)* | 1,401/12,107 (11.6) |
| Any abnormalities | 1/38 (2.6) | 0/26 (0) | 29/960 (3.0) | 40/1,703 (2.4) | 285/12,107 (2.4) |
| **BMI of 23–24.9 kg/m²** | | | | | |
| Gestational age at 22–30 weeks | 0/10 (0) | 0/11 (0) | 0/257 (0) | 1/354 (0.3) | 14/1,829 (0.8) |
| Gestational age at 31–36 weeks | 3/10 (30.0)* | 2/11 (18.2) | 25/257 (9.7)* | 22/354 (6.2) | 81/1,829 (4.4) |
| Caesarean section | 5/11 (45.5) | 6/12 (50.0) | 95/258 (36.8)* | 102/357 (28.6) | 466/1,833 (25.4) |
| Any labour complication | 4/11 (36.4) | 8/12 (66.7) | 256/258 (99.2)* | 170/357 (47.6) | 853/1,833 (46.5) |
| Any neonatal complication | 5/11 (45.5)* | 2/12 (16.7) | 49/258 (19.0)* | 60/357 (16.8) | 248/1,833 (13.5) |
| Any abnormalities | 2/11 (18.2)* | 1/12 (8.3) | 8/258 (3.1) | 7/357 (2.0) | 47/1,833 (2.6) |
| **BMI of ≥25 kg/m²** | | | | | |
| Gestational age at 22–30 weeks | 0/11 (0) | 3/55 (5.5)* | 7/550 (1.3) | 3/423 (0.7) | 9/1,517 (0.6) |
| Gestational age at 31–36 weeks | 1/11 (9.1) | 8/55 (14.6)* | 58/550 (10.6)* | 32/423 (7.6) | 88/1,517 (5.8) |
| Caesarean section | 3/11 (27.3) | 27/57 (47.4)* | 245/551 (44.5)* | 159/424 (37.5)* | 457/1,528 (29.9) |
| Any labour complication | 7/11 (63.6) | 37/57 (64.9)* | 547/551 (99.3)* | 205424 (48.4) | 693/1,528 (45.4) |
| Any neonatal complication | 4/11 (36.4) | 23/57 (40.4)* | 135/551 (24.5)* | 76/424 (17.9)* | 207/1,528 (13.6) |
| Any abnormalities | 0/11 (0) | 1/57 (1.8) | 29/551 (5.3)* | 17/424 (4.0) | 45/1,528 (3.0) |

Data are n (%).

*p<0.05 compared with the non-diabetes group (by Fisher's exact test).

BMI, body mass index; GDM, gestational diabetes.

placental weight was significantly higher in any of the diabetes and screened GDM groups than in the non-diabetes group (all p<0.05).

Table 6 shows gestational age and anthropometric measures in relation to the mothers' BMI. The proportion of caesarean section tended to be higher in mothers with type 1 or 2 diabetes and a BMI of ≤24.9 kg/m² compared with that in those without diabetes. The proportions of any complications of labour and any neonatal complications tended to be higher in mothers with screened GDM than in those without diabetes and a BMI of ≥18.5 kg/m².

## Discussion

In our study, we observed the following main findings. Birth weight in the type 1 diabetes group was the heaviest among the diabetes groups. Additionally, placental weight was significantly heavier in the type 1 diabetes, GDM, and screened GDM groups than in the non-diabetes group (Table 1). The rates of preterm birth, gestational hypertension, and neonatal jaundice were more frequent in the type 1 diabetes, type 2 diabetes, and screened GDM groups

than in the non-diabetes group (Table 2). In mothers with a positive result in the 50-g GCT during pregnancy (screened GDM group), the proportions of any complications of labour and any neonatal complications tended to be higher than those in mothers without diabetes (Tables 2 and 3).

The gestational age of neonates of mothers in the type 1 diabetes group was significantly shorter than that of neonates of mothers in the non-diabetes group. However, the birth weight of neonates of mothers in the type 1 diabetes group was heavier than that of neonates of mothers in the non-diabetes group (Tables 1 and 4). The head circumference in the type 1 diabetes group was similar to that in the non-diabetes group, but the chest circumference in the type 1 diabetes group was longer. This difference in birth weight between groups may be attributed to an increased amount of neonatal body fat, which was previously reported in delivery of mothers with diabetes [13]. In fact, the HAPO study, which was a cohort study that followed >25,000 pregnant women in nine countries, showed a strong correlation between maternal plasma glucose levels and foetal adiposity [14]. Higher foetal plasma glucose levels in mothers with diabetes can facilitate hyperplasia of the foetal pancreas [15]. Hyperplasia of the pancreas leads to an increased amount of insulin, corresponding to high plasma glucose levels [16]. An increased secretion of insulin can increase body fat. Consequently, heavier neonates could be born from mothers with type 1 diabetes.

In our study, placental weight of neonates of mothers with type 1 or 2 diabetes was heavy (Table 1). These results are supported by the data of maternal age (Table 4) and term and single births (Table 5). Previous data suggested that placental weight was heaviest in mothers with GDM, followed by those with non-normal glucose tolerance and those with normal glucose tolerance [17]. Placental weight can become heavier with the presence of maternal diabetes [18]. Insulin and glucose are involved in foetal and placental angiogenesis and vasculogenesis [19]. A previous study showed that a hypervascular placenta in GDM [20] was associated with intrauterine foetal growth retardation [21], which also appears in a diabetic status [22].

A large placenta and macrosomia may represent hyperleptinaemia, hyperinsulinaemia, and oxidative stress *in utero* [23]. Transportation of glucose to the foetus through the placenta does not depend upon the insulin receptor, but on glucose transporter type 1 [24, 25]. However, maternal insulin can activate the signalling pathway of placental insulin receptors [26], and therefore, the maternal diabetic status may affect placental metabolism [27]. Studies on the insulin signalling pathway in the placenta under the condition of diabetes are scarce. A few studies have suggested that insulin resistance can change placental transportation of nutrients [28, 29]. A large placenta with altered metabolism may delay foetal development, and complications and abnormalities may occur.

The proportion of macrosomia, which is frequent with the presence of a high body mass index [30] and diabetes [31], was highest (5.9%) in mothers with type 2 diabetes in our study (Table 2). A Japanese multicentre study that analysed data from 2003 to 2009 showed improved glycaemic control of mothers with type 1 or 2 diabetes [32]. In this previous report, the proportion of macrosomia was 4.6% in type 1 diabetes and 5.0% in type 2 diabetes. Although our study lacked glycaemic control data, our finding of 5.9% for the rate of macrosomia in type 2 diabetes suggests that the birth weight of neonates of mothers with type 2 diabetes was not well controlled in the 2010s. The high proportion of macrosomia may partly have caused the high proportion of caesarean section in the diabetic condition. Because neonates with macrosomia also have a risk of disease in their growth [33], further studies of how to control birth weight in type 2 diabetes is necessary.

We found that 7.4% of neonates had any congenital abnormality in mothers with type 1 diabetes, especially those associated with the heart (1.5%) (Table 2). The proportion of congenital abnormalities was 1.9% in type 2 diabetes, 3.8% in GDM, and 2.5% in screened GDM, while it was 2.5% in non-diabetes. A recent report showed that among Japanese mothers, 5.1% with

GDM had congenital abnormalities and 6.4% with overt diabetes in pregnancy had congenital abnormalities [34]. A meta-analysis of non-Japanese data showed almost no difference in the proportion of congenital abnormalities between types 1 and 2 diabetes [35]. The proportion of any abnormality in delivery with type 1 diabetes was 8.8% between 1999 and 2000 in The Netherlands [36]. Between 2002 and 2003 in the UK, the proportion of congenital abnormalities was 37.0% in type 1 diabetes and 12.8% in type 2 diabetes [37]. Data from 2003 to 2009 in Japan showed that the proportion of neonatal congenital malformations was 4.6% in type 1 diabetes and 4.1% in type 2 diabetes [32]. The proportion of neonatal congenital malformations in the type 1 diabetes group was high in our study. This finding may be due to different criteria of diagnosing abnormalities. In mothers with type 1 diabetes with the highest rate of congenital abnormalities, a high standard of medical care for mothers and their children is required.

The HAPO study showed associations of maternal plasma glucose levels with birth weight and the incidence of primary caesarean section [7]. Our findings of high proportions of macrosomia and caesarean section in mothers with diabetes (Table 2) are consistent with the results of the HAPO study. We also found a high incidence of labour and neonatal complications in mothers with diabetes. Premature rupture of the membranes and gestational hypertension were more frequent in mothers with type 1 or 2 diabetes than in those without diabetes. The incidence of neonatal complications was also high in all of the groups of mothers with diabetes. Particularly, jaundice with treatment was most frequent in mothers with type 1 diabetes. Furthermore, in the screened GDM group, a medium (middle-level) risk of labour and neonatal complications was observed.

The present results suggested that there was a slightly elevated risk of complications of labour, neonatal complications, and congenital abnormalities in Japanese mothers, even in those with a positive result in the screening test for GDM (Tables 2, 3, and 6). This risk included caesarean section, gestational hypertension, macrosomia, jaundice, and abnormalities of the heart and chromosomes (Table 2). Few studies have investigated such detailed complications and abnormalities among Asian mothers with varied types of diabetes. Although the 50-g GCT may be a burden for mothers, the results of screening are important for clinicians to predict the risks during pregnancy and at delivery.

This study has several limitations. First, because the questionnaire was not designed to investigate our study question, whether there was a history of diagnosed type 1 and type 2 diabetes depended on self-reporting by the mothers. By contrast, the diagnosed GDM groups and the screened GDM group were categorised by the obstetrician's response. Second, the glycaemic control indices of glycated haemoglobin and glycated albumin were not measured. The use of these indices could have enabled calculation of more detailed complication risks.

## Conclusions

Our study shows that birth weight is heavier in neonates of mothers with type 1 diabetes than in neonates of mothers without diabetes. Placental weight is heavier in mothers with all types of diabetes and non-normal glucose tolerance than in those without diabetes. These results indicate that types of diabetes can predict the probability of labour and neonatal complications in delivery. A positive result of a screening test for GDM suggests not only non-normal glucose tolerance, but also a higher risk of perinatal complications, than a negative result.

## Supporting information

**S1 Table. Relative risks of perinatal outcomes of mothers with type 1 diabetes, type 2 diabetes, gestational diabetes, and impaired glucose tolerance.**
(DOCX)

## Acknowledgments

We express our gratitude to all study participants and co-operating healthcare providers who supported the JECS. We thank Ellen Knapp, PhD, from Edanz for editing a draft of this manuscript.

The members of the JECS Group as of 2021 are as follows: Michihiro Kamijima (principal investigator, Nagoya City University, Nagoya, Japan; jecsen@nies.go.jp), Shin Yamazaki (National Institute for Environmental Studies, Tsukuba, Japan), Yukihiro Ohya (National Center for Child Health and Development, Tokyo, Japan), Reiko Kishi (Hokkaido University, Sapporo, Japan), Nobuo Yaegashi (Tohoku University, Sendai, Japan), Koichi Hashimoto (Fukushima Medical University, Fukushima, Japan), Chisato Mori (Chiba University, Chiba, Japan), Shuichi Ito (Yokohama City University, Yokohama, Japan), Hidekuni Inadera (University of Toyama, Toyama, Japan), Takeo Nakayama (Kyoto University, Kyoto, Japan), Hiroyasu Iso (Osaka University, Suita, Japan), Masayuki Shima (Hyogo College of Medicine, Nishinomiya, Japan), Hiroshige Nakamura (Tottori University, Yonago, Japan), Narufumi Suganuma (Kochi University, Nankoku, Japan), Koichi Kusuhara (University of Occupational and Environmental Health, Kitakyushu, Japan), and Takahiko Katoh (Kumamoto University, Kumamoto, Japan).

## Author Contributions

**Conceptualization:** Hiroshi Yokomichi, Mie Mochizuki.

**Formal analysis:** Hiroshi Yokomichi.

**Funding acquisition:** Zentaro Yamagata.

**Investigation:** Hiroshi Yokomichi, Mie Mochizuki.

**Methodology:** Hiroshi Yokomichi.

**Project administration:** Ryoji Shinohara, Megumi Kushima, Sayaka Horiuchi, Sanae Otawa, Zentaro Yamagata.

**Supervision:** Ryoji Shinohara, Zentaro Yamagata.

**Writing – original draft:** Hiroshi Yokomichi, Mie Mochizuki.

**Writing – review & editing:** Hiroshi Yokomichi, Mie Mochizuki, Ryoji Shinohara, Megumi Kushima, Sayaka Horiuchi, Reiji Kojima, Tadao Ooka, Yuka Akiyama, Kunio Miyake, Sanae Otawa, Zentaro Yamagata.

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
