## [Decision Letter · Decision Letter 0]

7 Apr 2022

PONE-D-22-06763Gestational age, birth weight and perinatal complications in mothers with diabetes and impaired glucose tolerance: Japan Environment and Children’s Study cohortPLOS ONE

Dear Dr. Yokomichi,

Thank you for submitting your manuscript to PLOS ONE. After careful consideration, we feel that it has merit but does not fully meet PLOS ONE’s publication criteria as it currently stands. Therefore, we invite you to submit a revised version of the manuscript that addresses the points raised during the review process.

 This is a well-written and interesting manuscript which both reviewers rate positively. One reviewer has suggested a minor revision, which I think is worth doing to improve the manuscript. Other than that:1. In the abstract please rephrase or define what is meant by "medium risk" (lines 32 & 35-36).2. Please provide more details about the 50g OGTT. Were the women fasted before the test? In what form was the glucose administered? When were blood samples collected? 3. Please provide a reference to the use of 140 mg/dL as the cut off for GDM in Japan (line 88). Which sample (e.g. fasting, 60 min.) was used to test against this criterion?4. Why was a Student's t-test used to analyse the data when there were more than two groups? How were data dealt with if they were not normally distributed or variances between the groups were different?

I look forward to seeing the revised manuscript. 

We look forward to receiving your revised manuscript.

Kind regards,

Clive J Petry, PhD

Academic Editor

PLOS ONE

Journal Requirements:

4. One of the noted authors is a group or consortium Japan Environment and Children’s Study Group. In addition to naming the author group, please list the individual authors and affiliations within this group in the acknowledgments section of your manuscript. Please also indicate clearly a lead author for this group along with a contact email address.

Reviewers' comments:

Reviewer's Responses to Questions

**Comments to the Author**

1. Is the manuscript technically sound, and do the data support the conclusions?

Reviewer #1: Yes

Reviewer #2: Yes

2. Has the statistical analysis been performed appropriately and rigorously? 

Reviewer #1: Yes

Reviewer #2: Yes

3. Have the authors made all data underlying the findings in their manuscript fully available?

Reviewer #1: Yes

Reviewer #2: Yes

4. Is the manuscript presented in an intelligible fashion and written in standard English?

Reviewer #1: Yes

Reviewer #2: Yes

5. Review Comments to the Author

Reviewer #1: Thanks for preparing this paper.

I would suggest to show Relative Risk ( RR ) with considering probable confounding factors , there is another published paper as reference

Uchinuma, H., Tsuchiya, K., Sekine, T. et al. Gestational body weight gain and risk of low birth weight or macrosomia in women of Japan: a nationwide cohort study. Int J Obes 45, 2666–2674 (2021). https://doi.org/10.1038/s41366-021-00947-7

Good luck

Reviewer #2: I was very impressed with the sample large size of your paper,. it was determined the perinatal risks risks in a large Japanese birth cohort analysing data of 27,855 neonate–mother pairs from 15 Regional Centres in The Japan Environment and Children’s Study cohort between March 2011 and November 2014. It was included 68 mothers with type 1 diabetes, 116 with type 2 diabetes, 735 with gestational diabetes (determined by questionnaire), and 4,384 with plasma glucose levels ≥140 mg/dL.(shown by a screening test for gestational diabetes).

Another aspects to be considered is the high quality of analysis and the English written.

6. PLOS authors have the option to publish the peer review history of their article (what does this mean?). If published, this will include your full peer review and any attached files.

Reviewer #1: No

Reviewer #2: No

---

## [Author Response · Author response to Decision Letter 0]

16 May 2022

Responses to the Editor and reviewers

As requested, we have prepared a revised version of our manuscript on the basis of the comments and suggestions received. We hope that these revisions have sufficiently addressed the Editor’s and reviewers’ concerns. Our point-by-point responses are included below each comment (in italics), with line numbers indicating the relevant changes in the revised manuscript. We extend our sincere thanks to the Editor and reviewers for all of the helpful comments provided. 

Responses to additional comments from the Editor

This is a well-written and interesting manuscript which both reviewers rate positively. One reviewer has suggested a minor revision, which I think is worth doing to improve the manuscript. 

1. In the abstract please rephrase or define what is meant by "medium risk" (lines 32 & 35-36).

We thank the Editor for the suggestion. To clarify what we mean by “medium risk”, we have added “middle-level” in parentheses. We have revised the main text accordingly. 

Line 36: A positive result in a screening test for gestational diabetes suggests not only a non-normal glucose tolerance, but also a medium (middle-level) risk of perinatal complications. 

Line 246: Furthermore, in the screened GDM group, a medium (middle-level) risk of labour and neonatal complications was observed.

2. Please provide more details about the 50g OGTT. Were the women fasted before the test? In what form was the glucose administered? When were blood samples collected? 

3. Please provide a reference to the use of 140 mg/dL as the cut off for GDM in Japan (line 88). Which sample (e.g. fasting, 60 min.) was used to test against this criterion?

We thank the reviewer for the comments. The 50-g oral glucose challenge test (GCT) is usually conducted for screening of gestational diabetes in pregnant women with a gestational age between 24 and 28 weeks, regardless of the mealtime. Plasma glucose levels ≥140 mg/dL one hour after the administration of 50 g of glucose is regarded as positive in the screening test. We have described the details of this test as follows. 

Line 50: In Japan, a 50-g oral glucose challenge test (GCT) in the second trimester of pregnancy is recommended for pregnant women without diabetes as screening [5]. If mothers have a positive result in the GCT, they should undergo detailed examinations [6]. 

References

5. Maegawa Y, Sugiyama T, Kusaka H, Mitao M, Toyoda N. Screening tests for gestational diabetes in Japan in the 1st and 2nd trimester of pregnancy. Diabetes Res Clin Pract. 2003;62:47-53. doi: 10.1016/s0168-8227(03)00146-3 PMID: 14581157

6. Miyakoshi K, Tanaka M, Ueno K, Uehara K, Ishimoto H, Yoshimura Y. Cutoff value of 1 h, 50 g glucose challenge test for screening of gestational diabetes mellitus in a Japanese population. Diabetes Res Clin Pract. 2003;60:63-7. doi: 10.1016/s0168-8227(02)00274-7 PMID: 12639767 

Line 83: Mothers answered about the experience of diagnosed type 1 diabetes, type 2 diabetes, and other endocrinological disease in the first trimester in the questionnaire. Data of mothers with endocrinological disease other than diabetes (e.g., Graves’ disease or Hashimoto disease) were excluded from the analysis. Obstetricians conducted a 50-g oral GCT for the pregnant participants at the second trimester. 

Lin 90: In Japan, GDM or overt diabetes in pregnancy is screened twice in all pregnant women. In the first trimester of pregnancy, random plasma glucose levels are evaluated with a cut-off value of 100 mg/dL or 95 mg/dL (the cut-off value depends on the institution). In the second trimester, the 50-g oral GCT with a cut-off value of 140 mg/dL at 1 hour or random plasma glucose levels with a cut-off value of 100 mg/dL is used for screening [5]. The GCT is usually conducted regardless of the mealtime. These screening tests are used to evaluate the glucose tolerance in pregnant women. If mothers have a positive result in the screening, they should undergo detailed examinations for determining the presence of GDM or overt diabetes in pregnancy [6].

Line 98: We categorised mothers in the groups of type 1 diabetes, type 2 diabetes, GDM, screened gestational diabetes mellitus (screened GDM), and non-diabetes as follows. First, we identified mothers with type 1 diabetes, type 2 diabetes, or GDM from the questionnaire. Overlapping of the types of diabetes was not permitted. Next, mothers with plasma glucose levels ≥140 mg/dL [6] at 1 hour in the GCT in the second trimester were categorised into the screened GDM group in this study. Although there were also data of GCT results in the third trimester, we did not use these data because the GCT is recommended to be conducted in the second trimester. Mothers without screened GDM, any type of diabetes, endocrinological disease, or plasma glucose levels ≥140 mg/dL at 1 hour in the GCT in the third trimester were categorised into the non-diabetes group. 

4. Why was a Student's t-test used to analyse the data when there were more than two groups? How were data dealt with if they were not normally distributed or variances between the groups were different? 

We thank the editor for the comment. We compared the mean values in a pairwise manner to determine the perinatal risk of any type of diabetes and elevated plasma glucose levels compared with those of the non-diabetes group. 

Student’s t-test is usually not conducted when observing normally distributed sample data, but in a population that is considered to be normally distributed. In this study, Welch’s t test was used with the assumption of unequal variances. We also consider that Welch’s t test can be used in the situation of equal variances. In a relatively large sample size, the t test is powerful, regardless of the distribution of the population. Therefore, we used the t test in this study. As the sample size in two groups becomes large, the t test is robust, even when x does not follow a normal distribution. The reason for this robustness is that the t test is based on the means of two groups. Because of the central limit theorem, the distribution of the sample means in repeated sampling converges to a normal distribution, irrespective of the distribution of x in the population. Additionally, the estimator that the t test uses for the standard error of the sample means is consistent, irrespective of the distribution of x. Therefore, this estimator is also unaffected by normality. We have added the following sentence.

Line 117: We conducted Welch’s t test or Fisher’s exact test for the comparison of variables between mothers with diabetes and those without diabetes.

Journal requirements

We thank the Editor for the suggestion. We have edited our manuscript to meet PLOS ONE’s style requirements. 

We have reviewed the references to ensure that they are complete and correct. 

7. We note that you have indicated that data from this study are available upon request. PLOS only allows data to be available upon request if there are legal or ethical restrictions on sharing data publicly. For more information on unacceptable data access restrictions, please see http://journals.plos.org/plosone/s/data-availability#loc-unacceptable-data-access-restrictions. 

We thank the Editor for this information. The data are owned by the Japan Environment and Children’s Study Group. The ethics committee of this group imposes restriction on data sharing. We have described contact information to which a data request could be sent. We have also described this information in our cover letter. 

8. One of the noted authors is a group or consortium Japan Environment and Children’s Study Group. In addition to naming the author group, please list the individual authors and affiliations within this group in the acknowledgments section of your manuscript. Please also indicate clearly a lead author for this group along with a contact email address.

We thank the Editor for the suggestion. We have added the names of the JECS group to the Acknowledgments section. We have also indicated the contact e-mail address of the principal investigator. 

Line 273: “The members of the JECS Group as of 2021 are as follows: Michihiro Kamijima (principal investigator, Nagoya City University, Nagoya, Japan; jecsen@nies.go.jp), Shin Yamazaki (National Institute for Environmental Studies, Tsukuba, Japan), Yukihiro Ohya (National Center for Child Health and Development, Tokyo, Japan), Reiko Kishi (Hokkaido University, Sapporo, Japan), Nobuo Yaegashi (Tohoku University, Sendai, Japan), Koichi Hashimoto (Fukushima Medical University, Fukushima, Japan), Chisato Mori (Chiba University, Chiba, Japan), Shuichi Ito (Yokohama City University, Yokohama, Japan), Hidekuni Inadera (University of Toyama, Toyama, Japan), Takeo Nakayama (Kyoto University, Kyoto, Japan), Hiroyasu Iso (Osaka University, Suita, Japan), Masayuki Shima (Hyogo College of Medicine, Nishinomiya, Japan), Hiroshige Nakamura (Tottori University, Yonago, Japan), Narufumi Suganuma (Kochi University, Nankoku, Japan), Koichi Kusuhara (University of Occupational and Environmental Health, Kitakyushu, Japan), and Takahiko Katoh (Kumamoto University, Kumamoto, Japan).” 

Responses to comments from reviewer #1

Thanks for preparing this paper.

9. I would suggest to show Relative Risk (RR) with considering probable confounding factors.

We thank the reviewer for the suggestion. We calculated the relative risks compared with those of the non-diabetes group, while taking into considering maternal age and maternal smoking status in the first trimester in pregnancy as confounding factors. We have added this information to the revised manuscript and to a table (new Table 3). 

Line 112: Poisson regression was used to calculate the relative risks of perinatal complications for type 1 diabetes, type 2 diabetes, GDM, and screened GDM compared with those for the non-diabetes group, with adjustment for maternal age and maternal smoking status in the first trimester in pregnancy.

10. There is another published paper as reference

Uchinuma, H., Tsuchiya, K., Sekine, T. et al. Gestational body weight gain and risk of low birth weight or macrosomia in women of Japan: a nationwide cohort study. Int J Obes 45, 2666–2674 (2021). https://doi.org/10.1038/s41366-021-00947-7. 

We thank the reviewer for the suggestion to add this reference. We have cited the reference as follows. 

Line 215: The proportion of macrosomia, which is frequent with the presence of a high body mass index [30] and diabetes [31], was highest (5.9%) in mothers with type 2 diabetes in our study (Table 2).

Reference

[30] Uchinuma H, Tsuchiya K, Sekine T, Horiuchi S, Kushima M, Otawa S, et al. Gestational body weight gain and risk of low birth weight or macrosomia in women of Japan: a nationwide cohort study. Int J Obesity. 2021;45:2666-74. doi: 10.1038/s41366-021-00947-7 PMID: 34465856 

Responses to comments from reviewer #2

I was very impressed with the sample large size of your paper. It was determined the perinatal risks in a large Japanese birth cohort analysing data of 27,855 neonate–mother pairs from 15 Regional Centres in The Japan Environment and Children’s Study cohort between March 2011 and November 2014. It was included 68 mothers with type 1 diabetes, 116 with type 2 diabetes, 735 with gestational diabetes (determined by questionnaire), and 4,384 with plasma glucose levels ≥140 mg/dL (shown by a screening test for gestational diabetes).

Another aspect to be considered is the high quality of analysis and the English written.

We appreciate the reviewer’s positive comment. We have added the relative risks of perinatal adverse outcomes for any type of diabetes, while taking into consideration maternal age and maternal smoking status in the first trimester in pregnancy, to new Table 3. We have also further polished the English language in the revised manuscript. 

Request for modifications in this revision from the authors

If this is permitted, we have revised the following two issues in this revision. 

11. We originally defined gestational diabetes from the questionnaire provided to mothers in the first trimester. However, we have since realised that most gestational diabetes is diagnosed from the second trimester. Because we have data of the diagnosis of gestational diabetes by obstetricians from the second to the third trimesters, we have replaced the definition of gestational diabetes in our study. We consider that this revised definition of gestational diabetes more precisely groups patients with gestational diabetes. We apologise for this change. 

12. The International Obesity Task Force recommends a cut-off of a BMI ≥23 kg/m2 for overweight and ≥25.0 kg/m2 for obesity in Asian people, according to the risks for type 2 diabetes and hypertension [1]. Therefore, we have changed the cut-off of the BMI from 22 kg/m2 to 23 kg/m2 in Table 6. 

Reference

[1] International Obesity Task Force, World Health Organization. The Asian-Pacific perspective: Redefining obesity and its treatment. Geneva: WHO Western Pacific Region, 2000.

We hope that our responses to the Editor’s and reviewers’ comments and the corresponding manuscript revisions have addressed the main issues raised. We are grateful for the helpful suggestions, and we hope that our manuscript is now suitable for publication in PLOS ONE.

Yours sincerely,

Hiroshi Yokomichi 

Department of Health Sciences, University of Yamanashi,

1110 Shimokato, Chuo, Yamanashi 409-3898, Japan 

E-mail: hyokomichi@yamanashi.ac.jp

Phone: +81 55 273 9569

Fax: +81 55 273 7882

---

## [Editor Report · Decision Letter 1]

25 May 2022

Gestational age, birth weight and perinatal complications in mothers with diabetes and impaired glucose tolerance: Japan Environment and Children’s Study cohort

PONE-D-22-06763R1

Dear Dr. Yokomichi,

We’re pleased to inform you that your manuscript has been judged scientifically suitable for publication and will be formally accepted for publication once it meets all outstanding technical requirements.

Kind regards,

Clive J. Petry, PhD

Academic Editor

PLOS ONE

Additional Editor Comments (optional):

Congratulations to all this authors - this is a very nice paper!
---

## [Editor Report · Acceptance letter]

27 May 2022

PONE-D-22-06763R1 

Gestational age, birth weight, and perinatal complications in mothers with diabetes and impaired glucose tolerance: Japan Environment and Children’s Study cohort 

Dear Dr. Yokomichi:

I'm pleased to inform you that your manuscript has been deemed suitable for publication in PLOS ONE. Congratulations! Your manuscript is now with our production department. 

Kind regards, 

on behalf of

Dr. Clive J. Petry 

Academic Editor

PLOS ONE